# Effectiveness of Nature-Based Interventions in Reducing Agitation Among Older Adults with Dementia: A Systematic Review and Meta-Analysis

**DOI:** 10.3390/healthcare13141727

**Published:** 2025-07-17

**Authors:** Eun Yeong Choe, Jennifer Yoohyun Lee, Jed Montayre

**Affiliations:** 1School of Design, The Hong Kong Polytechnic University, Hong Kong; jenn-y.lee@polyu.edu.hk; 2School of Nursing, The Hong Kong Polytechnic University, Hong Kong; jed-ray.montayre@polyu.edu.hk

**Keywords:** nature stimuli, sensory experience, biophilic design, quality of life, long-term residential care

## Abstract

**Background/Objectives**: The role of environmental modifications and design in mitigating behavioural symptoms is increasingly being recognised as a way to address the psychosocial needs of individuals with dementia. This study aims to investigate various nature-based interventions for reducing agitation in people with dementia in long-term residential care environments. **Methods**: Database searches were conducted on MEDLINE, PsycINFO, Scopus, and Web of Science. A literature search was conducted with the following inclusion criteria: (i) peer-reviewed journal publication written in English; (ii) random controlled trials (RCTs) and quasi-experimental design with results for pre- and post-testing reported; (iii) interventions using natural elements, where the effectiveness of the reduction in agitation was measured using a validated instrument; and (iv) participants aged 65 and older with dementia residing in long-term care facilities. **Results**: This meta-analysis included 29 studies with 733 participants. The results showed that such interventions had a significant negative mean effect on lowering agitation in this population. Additionally, intervention settings (indoor vs. outdoor) and the presence of social interaction were significant predictors of the effect size for agitation reduction. At the same time, no significant differences in effect size were observed between the types of experiences with nature (indirect vs. direct) or the duration of the interventions. **Conclusions**: This study demonstrates that, when thoughtfully applied, nature-based interventions can significantly alleviate agitation in patients with dementia residing in long-term residential care facilities. This review lays the groundwork for future research aimed at developing design guidelines and planning strategies to integrate natural elements into dementia-friendly environments effectively.

## 1. Introduction

As a person’s dementia progresses, they may experience agitation, psychosis, anxiety, depression, and apathy [1]. The behavioural symptoms of dementia may also include patients’ inappropriate physical and verbal actions towards family members and caregivers. Due to the challenging and disruptive behaviours that characterise dementia, nearly every person with dementia is eventually placed in a nursing home at some point during the illness [2]. However, behavioural symptoms are often exacerbated or triggered by a mismatch between individuals’ needs or excessive stressors and the environment [3]. Therefore, modifying the environment to accommodate the needs of dementia patients could be an important strategy to mitigate agitation and other behavioural symptoms.

To date, a number of studies have examined and reviewed the relationship between people with dementia and their environments. For example, a systematic review by Marquardt et al. [4] categorised 169 studies into four main areas: basic design decisions, environmental attributes, ambience, and environmental information, and illustrated how these design interventions impact the behaviours, functions, well-being, social engagement, and orientations of people with dementia. Chaudhury et al. [5] also reviewed 103 studies and identified the influence of several factors, including environmental characteristics of social spaces, unit size, spatial layout, homelike character, and sensory stimulation, on residents’ behaviours and well-being in care facilities. Furthermore, Barrett et al. [6] provided a synthesis of diverse evidence leading to the formulation of three design principles (manageable cognitive load, clear sequencing, and an appropriate level of stimulation), backed up by more in-depth design parameters. The findings from the reviews indicated that the environment could have a therapeutic effect on people with dementia, helping them improve and preserve their well-being, behaviour, functionality, and independence. In particular, the reviews have shown that sensory stimulation approaches, which are abundant in nature, can serve as effective interventions. Although individuals with dementia respond differently to natural stimuli, studies have shown certain benefits, including reduced agitation and depression, improved mood, positive interactions with staff, and enhanced well-being [7,8,9]. Therefore, designing interventions and environments that provide opportunities for people with dementia to interact with nature could be crucial in promoting their health, well-being, and quality of life.

The design discourse has increasingly recognised and promoted those physical living environments with natural elements as essentially therapeutic. For example, Wilson [10] introduced the term ‘biophilia’, which describes the inherent connection humans have with nature. When individuals encounter nonthreatening natural stimuli, their emotional state improves, resulting in positive changes in physiological activity, enhanced attention and memory, and reduced negative emotions [11,12,13]. For individuals with dementia, nature offers a stimulating environment with sensory experiences and opportunities for engaging in social activities, which in turn have a positive impact on their health and well-being. A qualitative review by Orr et al. [14] examined how older people with dementia described their sensory experiences and connectedness with the natural world. The authors reported that older adults found significant pleasure and enjoyment from observing and interacting with nature and that these activities appeared to have therapeutic effects on their mood and quality of life. Similarly, Bennett et al. [15] found that continued engagement with outdoor activities improved associated support for self and identity among people with dementia. In their review of 16 studies on sensory gardens and horticultural activities with participants with dementia living in care homes, Gonzalez and Kirkevold [16] reported that nature-based activities reduced agitation and restlessness, increased social engagement, and improved the sleep and well-being of people with dementia. Murroni et al. [17] also reviewed 16 quantitative studies on the benefits that garden visits and gardening therapy have on people with dementia. The review revealed that the most significant effects were engagement, agitation, stress, positive emotions, and improved medication use. Motealleh’s narrative review [18] of the effectiveness of outdoor natural landscapes focused on the behavioural and psychological symptoms of dementia, including agitation, apathy, and engagement. Furthermore, some studies have focused on specific natural elements within designs, such as images of nature, light, sound, and smell—also known as ‘biophilic design’. As an application of the concept of ‘biophilia’, biophilic design offers a sustainable design strategy that fosters connections between humans and nature in built environments [19]. This design concept involves direct experiences of nature, such as natural light, air, plants, animals, and water; indirect experiences of nature, including images of nature, natural materials, and naturalistic shapes and forms; and the simulation of light and smell, all of which evoke and mimic natural elements. A review of Goudriaan et al. [20] examined the influence of daylight and indoor lighting on older adults with dementia in care facilities. Although they found no conclusive evidence to suggest that indoor light reduces challenging behaviours, positive effects were observed in improving depressive symptoms and quality of life. Another review by D’Andrea et al. [21] synthesised the evidence regarding the efficacy of interventions involving olfactory stimulation (e.g., lavender or orange) for individuals with dementia. They demonstrated that exposure to scents can trigger memories of personal past experiences and autobiographical memory, thereby evoking positive emotions. More recently, Huang and Yuan [22] examined 11 studies on the health benefits aromatic scents have on older people with dementia. They indicated that herbs and aromatic plants had comprehensive health benefits, such as reduced agitated behaviour and enhanced cognitive functions and overall well-being.

Nevertheless, a key finding of the abovementioned reviews is the limited empirical evidence identifying the beneficial effects of nature-based applications on people with dementia. To date, insufficient attention has been given to the application of design and its role in enhancing long-term residential care environments. Furthermore, there is still a lack of systematic understanding of how nature-based interventions affect specific symptoms in long-term residents with dementia in their care settings. To address this research gap, this paper builds on earlier reviews and analyses and examines experimental studies that assess the impact of nature-based interventions on agitation in people with dementia. The focus is on studies involving interventions that aim to enhance sensory experiences and reduce agitation by utilising natural elements in long-term residential care environments. For this review, agitation is defined as ‘a response to either internal or external stimuli or both, described by behavioural or verbal disruptiveness, inappropriateness, aggressiveness and repetitiveness’ [23]. To systematically explore the relative impact of nature-based interventions on reducing residents’ agitation, the effects of the intervention settings (indoor vs. outdoor), the presence of social interaction, the types of experiences with nature (indirect vs. direct), and the duration of the interventions are examined. This review and meta-analysis focus on studies conducted in long-term residential care settings, aiming to serve as a valuable resource for those seeking to advance knowledge in this field and to identify effective dementia care practices across various contexts. Based on the conclusions drawn from existing studies, a research and practice agenda is proposed.

## 2. Materials and Methods

### 2.1. Inclusion Criteria

A literature search was conducted with the following inclusion criteria: (i) peer-reviewed journal publications written in English; (ii) random controlled trials (RCTs) and quasi-experimental design with results for pre- and post-testing reported; (iii) interventions using natural elements, where the effectiveness of the reduction in agitation was measured using a validated instrument, including the Cohen–Mansfield Agitation Inventory (CMAI), Agitated Behaviour Scale (ABS), and Neuropsychiatric Inventory (NPI); and (iv) participants aged 65 and older with dementia residing in long-term care facilities. Furthermore, following Kellert and Calabrese’s key principles of biophilic design, only the studies that examined interventions involving ‘repeated and sustained engagement with nature’ [19] were considered.

### 2.2. Search Strategy

This meta-analysis adhered to the Preferred Reporting Items for Systematic Reviews and Meta-Analyses (PRISMAs) guidelines. Database searches were conducted on MEDLINE, PsycINFO, Scopus, and Web of Science. The final search was completed in July 2024. Four sets of filters were applied to the database search, all of which were set to search for terms in the titles or abstracts. The first filter was used to identify studies about natural elements: Title (natur* OR garden* OR green* OR farm* OR plant* OR biophilic OR sound OR light OR scent OR olfactory). The second filter was applied to identify studies on the dementia population: Abstract (dementia), whilst the third filter was used to identify studies on built environments: Abstract (design OR environment* OR setting). Additional papers were identified through manual searches of review papers and Google searches.

### 2.3. Data Collection Process and Data Items

The data comprised (i) the basic study information (author, year of publication, and country), (ii) study design, (iii) the sample information (number and mean age), (iv) intervention details (intervention content and duration), and (v) outcome measurement.

### 2.4. Quality Assessment of Eligible Studies

Two independent reviewers assessed all the studies meeting the inclusion criteria for external and internal validity using the JBI standardised appraisal tools for RCTs and quasi-experimental studies (see Appendix A). The questions in the appraisal tool were rated ‘Yes’, ‘No’, ‘Unclear’, or ‘Not applicable’. The overall methodological quality of each study was then calculated by adding all ‘Yes’ ratings and dividing them by the number of applicable questions to obtain a percentage [24]. The studies were rated ‘low methodological quality’, ‘adequate’, ‘moderate’ and ‘strong’ if they received a rating of less than 50%, between 50% and 69%, between 70% and 85%, and between 86% and 100%, respectively. Given that studies of ‘low methodological quality’ may compromise the quality of systematic review practice recommendations, all studies with such a rating were excluded.

### 2.5. Data Synthesis and Analysis

The software package Meta Essentials (www.erim.eur.nl/research-support (accessed on 5 April 2025) was used to analyse the data. Meta Essentials is a useful tool that automatically calculates effect sizes from different types of data and supports various meta-analysis tasks, such as subgroup analysis, moderator analysis, and checking for publication bias [25]. All outcomes included in the analyses were continuous. The extracted data included pre—post mean values and Cohen’s d and Hedges’ g effect sizes, as well as mean change scores, including the standard deviation (SD) and/or confidence interval (CI) values. A key eligibility criterion for inclusion was that studies must include a specific measure of agitation, which could be any of the validated measures with the available constituent subscales. Where missing data were found, the corresponding authors were contacted to request further information. In cases where the correlation coefficient (r) was not provided or available, data from other sources were used to estimate this correlation and a sensitivity analysis was conducted to determine how this affected the results.

Due to the variation in the samples analysed, the random effects model was chosen as the most appropriate analytical approach. Uncertainty about the findings was expressed in the form of a 95% CI. This was followed by the evaluation of effect sizes [26], with 0.2, 0.5, and 0.8 representing small, medium, and large effects, respectively. Heterogeneity amongst effect sizes was determined by a significant Q value (*p* < 0.10). The I^2^ statistic is indicative of the degree of variability in effect sizes, with the ranges of 1–49, 50–74, and 75–100 signifying low, moderate, and high heterogeneity, respectively.

Moreover, in instances of high heterogeneity, three subgroup analyses were performed to assess whether (i) the intervention settings (indoor vs. outdoor), (ii) the presence of social interaction, (iii) the types of experiences with nature (indirect vs. direct), and (iv) the duration of interventions contributed to the variability in the effects. Finally, publication bias was examined using funnel plots and associated statistics. Assessing the impact of publication bias involved estimating the effect if the bias were absent using trim-and-fill analysis [27]. This method eliminated the most extreme effect sizes from the funnel plot and subsequently recalculated the effect size to enhance the symmetry of the funnel plots around the newly suggested effect size.

### 2.6. Sensitivity Analysis

Sensitivity analyses were applied to examine the robustness of the synthesised results. Following the guidance provided by Higgins and Green [28], in cases where there was an estimated substantial or considerable risk of heterogeneity, further subgroup and/or sensitivity analysis was conducted and reported alongside the main analysis to provide more detailed insights. Subgroup analysis involved stratifying the findings by the intensity of the intervention and, where appropriate, reporting results only for the studies that were assessed as having high or moderate methodological quality. Furthermore, in instances where forest plots indicated clear outliers within the sample, the removal of these items from the main model was considered, after which the resulting change in the overall main effect and I^2^ values was reported.

## 3. Results

### 3.1. Study Selection

Our database search protocol identified 1193 citations. After removing 594 duplicated items, our initial pool consisted of 599 publications. Of these, 507 articles were excluded because they did not meet the inclusion criteria. The list was further refined to 92 entries by discarding those that were irrelevant. A total of 35 articles were excluded as they did not contain the statistical information needed for our analyses—it was not available in the published document and could not be obtained from the authors. A total of 57 studies underwent a full-text analysis. Studies that lacked essential statistical information (e.g., means, standard deviations, or sample sizes), presented data in incompatible formats, or were identified as having significant methodological flaws were excluded. This resulted in 31 articles in the final publication pool. Figure 1 presents the PRISMA flowchart.

### 3.2. Study Characteristics

The initial analysis included 31 studies (11 RCTs and 20 quasi-experimental studies) with 826 participants. Nine papers were published before 2009, 13 between 2010 and 2019, and 9 between 2020 and 2023. In terms of study design, 11 studies used a randomised controlled design, and 20 studies used a quasi-experimental design. This increased rate of publication illustrates the establishment and acceptance of nature as a supportive design intervention for older people with dementia residing in long-term residential care environments. The studies were conducted in such care environments in the following different countries: 11 studies were carried out in North America, 8 studies in Asia, 7 studies in Europe, and 5 studies in Australia. The mean age of participants ranged from 71.14 to 86.90 years. In terms of interventions, 7 studies used naturalistic lighting for circadian stimulus, 10 studies used olfactory stimulation, 4 studies used the sound of nature, and 10 studies used indoor and outdoor gardens. In terms of outcome measures, the most widely used scale to measure the participants’ agitation was the CMAI (n = 23), followed by the NPI (n = 7) and the ABS (n = 1). There was a wide range in the duration over which the intervention was delivered, ranging between just 3 days and several months. Only four studies included follow-up measures at 3 weeks [18], 4 weeks [29,30], and 6 weeks [31] upon completion of the study. The detailed characteristics of each study are summarised in Table 1.

### 3.3. Immediate Effects of Interventions

The total sample size for interventions assessing nature connectedness pre- and post-intervention was 826 (range 5–82) from 31 studies. A large negative mean effect was observed (g = −1.04 [95% CI −1.43, −0.65]), which was significant (*p* < 0.001). The heterogeneity of effects was also significant (Q = 324.56, *p* < 0.001) and had high inconsistency (I^2^ = 90.76%).

Through the sensitivity analysis, inadequate studies were systematically removed to improve statistical consistency, resulting in 29 remaining studies. Two studies were identified as outliers due to effect sizes being substantially different from the others. Accordingly, the total sample size for interventions assessing agitation pre- and post-intervention was 733 (range 5–82) from 29 studies. A large negative mean effect was observed (g = −0.86 [95% CI −1.15, −0.57]), which was significant (*p* < 0.001). The heterogeneity of effects was also significant (Q = 215.61, *p* < 0.001) and had a moderate inconsistency (I^2^ = 87.01%). Subgroup analyses were then conducted on 29 studies.

### 3.4. Subgroup Analyses

Subgroup analyses were carried out to investigate the factors affecting the effectiveness of natural contact and engagement on the following nature connection dimensions: (i) intervention settings (indoor vs. outdoor), (ii) the presence of social interaction, (iii) the types of experiences with nature (indirect vs. direct), and (iv) the duration of interventions.

#### 3.4.1. Intervention Settings

Twenty studies were coded as indoor interventions and eleven as outdoor interventions. In this review, only outdoor interventions within care facilities (e.g., outdoor gardens directly accessible from living areas) were included to minimise selection bias. The test for subgroup differences suggested that there was a statistically significant subgroup effect (*p* = 0.040), indicating that the intervention settings statistically significantly affected the effectiveness of the interventions. For indoor settings, a large effect was observed (g = −1.00 [95% CI −1.38, −0.62], I^2^ = 90.34%), whilst a medium effect was found for outdoor settings (g = −0.76 [95% CI −1.19, −0.33], I^2^ = 49.38%), as shown in Figure 2.

#### 3.4.2. The Presence of Social Interaction

For the subgroup analysis, studies were categorised based on whether the interventions applied involved social interaction. The majority of the studies (n = 23) were individual interventions, whilst only 6 involved social interaction. The studies coded as ‘social interaction’ involved various interventions and activities to encourage social contact with caregivers, friends, or family members in natural environments. Those studies that required minimal contact to set up the intervention, such as applying oil to the skin or accompanying someone from the bedroom to the garden, were coded as ‘individual interventions’. The test for subgroup differences indicated a statistically significant subgroup effect (*p* = 0.028), indicating that the presence of social interaction statistically significantly modified the effect of the interventions. For the individual interventions, a medium effect (g = −0.73 [95% CI −1.02, −0.44], I^2^ = 85.44%) was found, whilst the interventions involving social interaction provided more significant outcomes (g = −0.99 [95% CI −1.59, −0.78], I^2^ = 68.62%), as shown in Figure 3.

#### 3.4.3. Types of Experiences with Nature

Following Richardson and Butler [57], indirect experiences of nature were identified in 18 studies, which involved contact with images of nature and natural materials (e.g., recorded sounds and smells) that evoked and mimicked nature. Furthermore, 11 studies were coded as ‘direct experiences of nature’, where participants engaged with natural environmental features, including air, natural light, vegetation, water bodies, landscapes, and ecosystems. The test for subgroup differences suggested that there was no statistically significant subgroup effect (*p* = 0.300), indicating that the types of experiences with nature did not modify the effect of the interventions (Figure 4).

#### 3.4.4. Duration of Interventions

Studies were categorised based on the duration of the intervention. The studies examined the impact of the intervention on participants’ agitation levels over periods ranging from 3 days to 12 months. The durations of 17 studies were less than 1 month, those of 7 studies were between 1 and 6 months, and those of 5 studies lasted more than 6 months. The test for subgroup differences suggested that there was no statistically significant subgroup effect (*p* = 0.970), indicating that the duration of intervention did not modify the effect of the interventions (Figure 5).

## 4. Discussion

This meta-analysis included 29 studies with 733 participants to investigate the application of various nature-based strategies for reducing agitation in people with dementia housed in long-term residential care environments. Overall, it was found that such interventions had a significant effect on reducing agitation in people with dementia. Furthermore, intervention settings (indoor vs. outdoor) and the presence of social interaction were significant predictors of the effect size for agitation reduction. No effect size differences were observed between the types of experiences with nature (indirect vs. direct) or the duration of interventions. However, only a few studies measured sustained changes. With only 4 out of 29 studies available, it is not possible to draw firm conclusions about the longevity of the effect on agitation.

### 4.1. The Effect of Intervention Settings

The results suggested that indoor interventions were more effective than outdoor interventions in reducing agitation in long-term residential care environments. Most interventions involving circadian or olfactory stimulation were conducted inside care homes, such as the participants’ bedrooms or in common areas (e.g., dining room). While Figueiro et al. [29] installed luminaires in a subject’s room for the lighting intervention, Saidane et al. [36] and Figueiro et al. [34] placed the floor lamps in both the participants’ bedrooms and common areas for those who spent most of the day outside their bedrooms. No differences in effects were observed between different indoor locations.

In recent years, there has been an increasing interest in incorporating outdoor gardens or patios in nursing facilities. Whilst previous studies have shown the benefits of multisensory experiences [14,16], there is less agreement regarding the value of the provision of outdoor spaces. A review by Fleming and Purandare [58] revealed that people with dementia have difficulty dealing with high levels of stimulation. They also tend to have a reduced ability to screen out unwanted stimuli and can easily become more confused, anxious, and agitated when overstimulated. Thus, for these individuals, there is a need to control sensory stimulation to avoid the reverse effect [4].

Moreover, Barrett et al. [6] proposed a model showing the links between design principles and the negative symptoms of dementia. The model indicated that moderate levels of stimulation would contribute to reduced apathy and agitation, whilst excessive stimulation was likely to lead to agitation and aggression. Thus, the authors suggested that environmental design should seek to find a balance where the individual remains engaged without being overwhelmed, making them stimulated, but not disturbed. Additionally, several studies have indicated barriers to the use of outdoor spaces. A lack of safety features (e.g., no handrails or poor surface materials), insufficient resting spaces, limited access to bathroom facilities and drinking fountains, and weather-related issues in outdoor environments could also diminish the benefits of natural experiences [3]. Thus, careful optimisation of stimulation levels and the provision of proper facilities are crucial to enhancing the positive effects of outdoor natural environments for people with dementia residing in long-term residential care.

### 4.2. The Effect of Social Interaction

It was found that nature-based interventions involving social interaction were more effective at reducing agitation than those that did not. In this study, the research identified as ‘social interaction’ involved the interventions designed to promote social contact with caregivers, friends, or family members in natural environments. For example, Edwards et al. [56] studied the impact of outdoor gardens as social spaces for a wide range of activities where individuals can meet friends and family members. Connell et al. [37] investigated the effects of 1 h recreational activities performed in small groups of 4–6 people to help them receive bright light exposure. The study of Whall et al. [48] also involved interactions between residents and staff during showers with natural simulations.

Whilst these studies demonstrated that social interaction might be a useful tool for enhancing the impact of nature-based applications in people with dementia, much uncertainty still exists regarding the impact of interactions with people on agitation in people with dementia. For instance, Burgio et al. [47] showed that staff touch and verbal interaction elicit agitation in some subjects but reduce it for others. Kolanowski and Litaker [59] observed that individuals who were more extraverted demonstrated significantly less agitation, whilst more introverted individuals showed greater agitation during high levels of social interaction. Thus, responses to social interactions seem to vary on an individual basis. Therefore, monitoring social interaction is crucial, as insufficient interaction can result in functional decline, whilst intensive human interaction may cause increased agitation. In this regard, further research is needed to identify the most effective methods for facilitating social interactions among residents without causing agitation.

This meta-analysis also tested the relative effects of the types of experiences with nature (indirect vs. direct) and the duration of interventions. For subgroup analysis, some studies identified indirect experiences of nature through contact with images and materials that evoked and mimicked nature. For example, Lin et al. [39] placed aroma diffusers near the participant’s bedside during nighttime sleep. In contrast, other studies were identified as ‘direct experiences of nature’, in which participants actively engaged with natural environmental features. Connell et al. [37] conducted the recreational program with small groups of 4–6 people in outdoor spaces to expose participants to natural light. However, the test for subgroup differences did not show a statistically significant subgroup effect. Moreover, the subgroup analysis of the duration of interventions also found no significant difference between subgroups (i.e., 1 month vs. 1–6 months vs. more than 6 months). Although some studies [33,37,39] were carried out over less than 1 month, the effects were large. In this review, coding experiences with nature are particularly challenging, as publications are often unable to capture the full scope of activities in which participants engage. Given that the focus was on the reported study methodology rather than on the participants’ actual experiences, this categorisation should be approached with caution.

### 4.3. Limitations of This Study

This study has potential limitations. The effect estimates in this analysis are based on the existing interventional studies. Differences in study designs, participant characteristics, and intervention types could affect the consistency of the results. Moreover, the scope of subgroup analyses was limited due to insufficient data or incomplete reporting, which restricted the ability to explore factors influencing the intervention’s effectiveness. Notwithstanding these limitations, this study provides some insights into the importance of considering nature-based interventions as an acceptable therapeutic intervention for agitation, particularly in efforts to enhance their environmental and social settings to cater to the needs of people with dementia.

### 4.4. Recommendations for Future Research and Practice

It is now widely recognised that providing environmental support can mitigate behavioural symptoms and improve the quality of life of persons with dementia by addressing their psychosocial needs. Several studies have shown the benefits of natural stimuli, but these have also indicated that individuals with dementia respond differently based on their cognitive load capacities and degrees of stimulation [6]. Getting to know the individual is crucial for identifying the underlying factors that trigger certain behaviours so that the potential for them to occur can be eliminated or minimised [60]. Therefore, further research is necessary to fully comprehend individual differences in the impact of biophilic design applications in care facilities. It involves understanding the environmental preferences, experiences, and activities that people with dementia respond to, which in turn promotes these aspects of health and helps them achieve a good quality of life.

Moreover, there has been little agreement on which optimum level of stimulation can reduce agitation in people with dementia. This review showed different outcomes in environmental stimulation (indoor vs. outdoor) and social interaction. In particular, relatively greater environmental stimulation had a negative effect on the benefits of nature-based applications, whilst human interaction had a positive effect on their benefits. These findings reveal the difficulty in distinguishing the contribution of the physical environment from that of social contact and engagement. Thus, further investigations and experimentation into the optimum level of stimulation for reducing agitation in people with dementia are strongly recommended.

More broadly, research is also needed to explore the combination of proper environmental stimulation with other care practice manipulations. Some studies that combine reduced stimulation with other manipulations have been shown to lessen behavioural disturbances [61]. For example, interventions in nursing homes aimed at reducing night-time noise have not succeeded in improving sleep duration, whereas a combination of increased daytime physical activity and decreased night-time noise levels can help alleviate agitation in nursing home residents [62,63]. A further study could assess the impact of combining nonpharmacological interventions with natural elements on agitation or other behavioural symptoms in older people with dementia. Clinicians and caregivers might also need some targeted education and training on the impact of nature-based interventions, which can be tailored to suit the care planning and management needs of behavioural symptoms in dementia. Finally, healthcare professionals working in facilities providing dementia care services could use nature-based interventions when undertaking environmental modifications to reduce risks related to falls and other health outcomes.

## 5. Conclusions

This study is among the first to conduct a meta-analysis of nature-based interventions as a strategy to reduce agitation in older people with dementia in institutional settings. This review shows the importance of considering nature-based applications as an acceptable therapeutic intervention for agitation in efforts to modify environments and enhance their environmental features to cater to the needs of people with dementia. More importantly, this work highlights that, when thoughtfully applied, natural elements can significantly alleviate agitation in dementia patients living in long-term residential care facilities. The intervention setting and the presence of social interaction were significant factors influencing the effectiveness of the intervention in reducing agitation. This review lays the groundwork for future research aimed at developing design guidelines and planning strategies to integrate natural elements into dementia-friendly environments.

## Figures and Tables

**Figure 1 healthcare-13-01727-f001:**
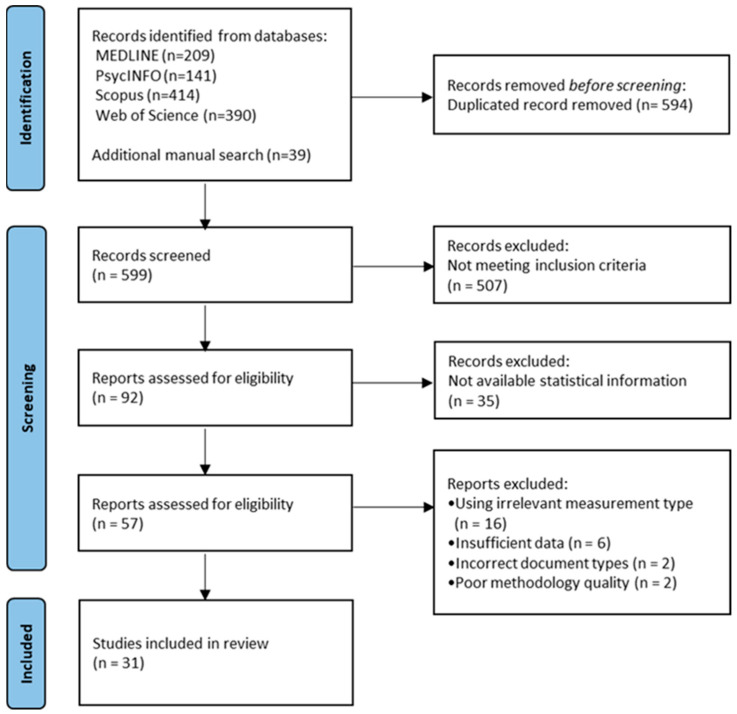
PRISMA flowchart of study selection.

**Figure 2 healthcare-13-01727-f002:**
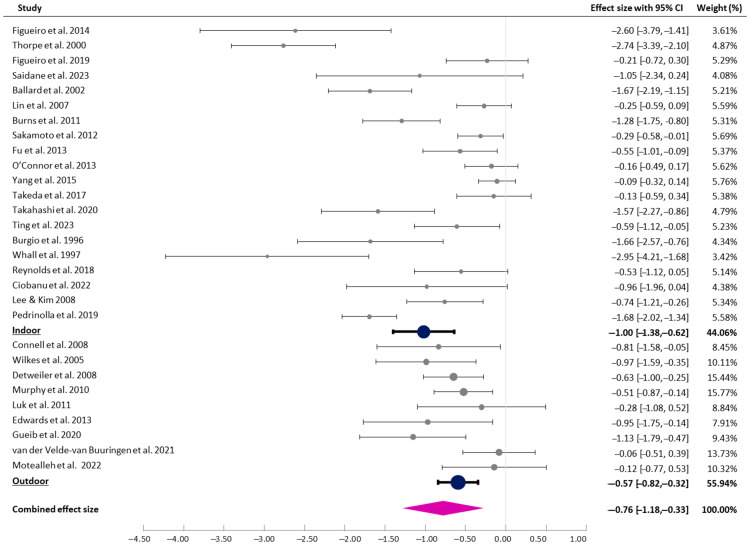
Forest plot of pre–post effects and CIs (indoor vs. outdoor interventions) [7,8,9,29,30,31,32,33,36,37,38,39,40,41,42,43,44,45,46,47,48,49,50,51,52,53,54,55,56].

**Figure 3 healthcare-13-01727-f003:**
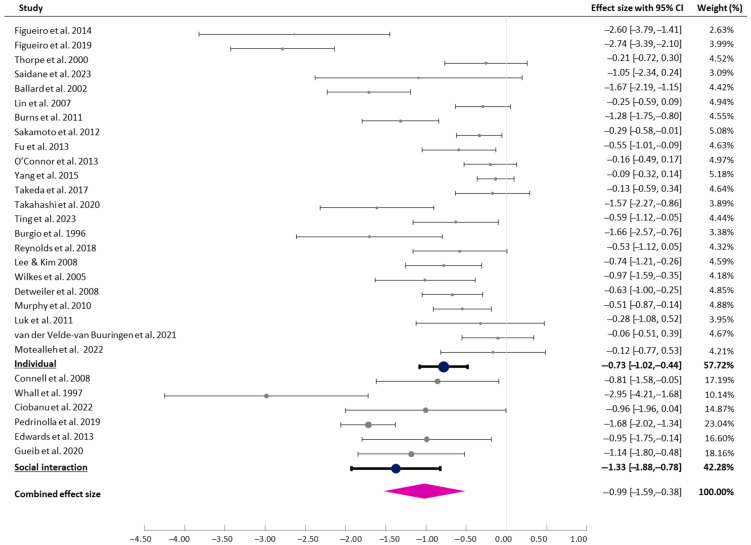
Forest plot of pre–post effects and CIs (individual vs. social interaction) [7,8,9,29,30,31,32,33,36,37,38,39,40,41,42,43,44,45,46,47,48,49,50,51,52,53,54,55,56].

**Figure 4 healthcare-13-01727-f004:**
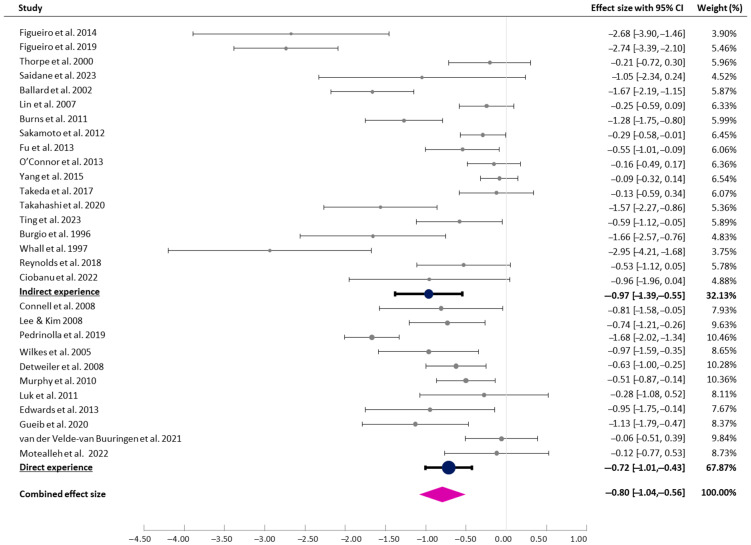
Forest plot of pre–post effects and CIs (indirect vs. direct experiences) [7,8,9,29,30,31,32,33,36,37,38,39,40,41,42,43,44,45,46,47,48,49,50,51,52,53,54,55,56].

**Figure 5 healthcare-13-01727-f005:**
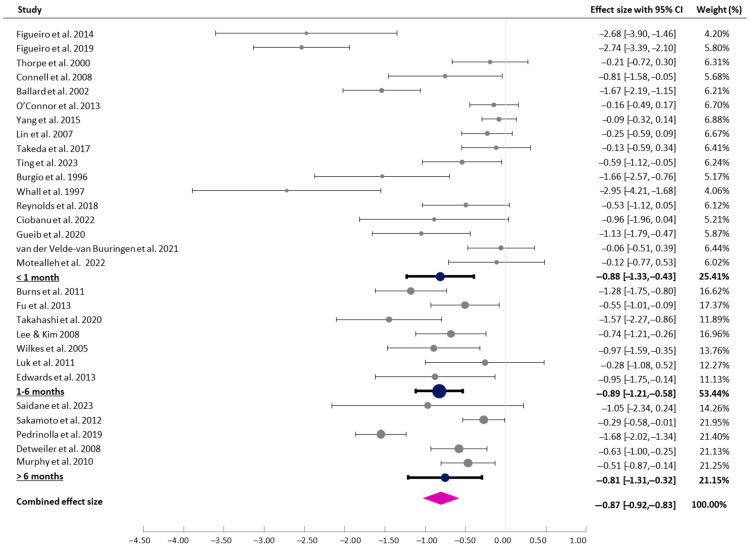
Forest plot of pre–post effects and CIs (duration of intervention) [7,8,9,29,30,31,32,33,36,37,38,39,40,41,42,43,44,45,46,47,48,49,50,51,52,53,54,55,56].

**Table 1 healthcare-13-01727-t001:** Characteristics of the studies included in the meta-analysis.

Author	Country	Study Design	Sample	Intervention	Duration	Scale Used
Figueiro et al. (2014) [29]	USA	Quasi-experimental design	n = 14, mean age 86.9 ± 4.4	Naturalistic lighting (Circadian Stimulus)	4 weeks	CMAI
Figueiro et al. (2019) [32]	USA	RCT	n = 46, mean age 85.1 ± 7.1	Naturalistic lighting (Circadian Stimulus)	4 weeks	CMAI
Thorpe et al. (2000) [33]	Canada	Quasi-experimental design	n = 16, mean age > 65	Naturalistic lighting (Circadian Stimulus)	1 week	CMAI
Figueiro et al. (2020) [34]	USA	Quasi-experimental design	n = 47, mean age 85.3 ± 7.1	Naturalistic lighting (Circadian Stimulus)	25 weeks	CMAI
Figueiro and Kales (2021) [35]	USA	Quasi-experimental design	n = 46, mean age 85.1 ± 7.1	Naturalistic lighting (Circadian Stimulus)	4 weeks	CMAI
Saidane et al. (2023) [36]	Denmark	Quasi-experimental design	n = 5, mean age > 65	Naturalistic lighting (Circadian Stimulus)	6 months	CMAI
Connell, Sanford and Lewis (2007) [37]	USA	Quasi-experimental design	n = 10, mean age 79.7 ± 8.3	Bright light exposure (via outdoor)	10 days	CMAI
Ballard et al. (2002) [38]	UK	RCT	n = 36, mean age 78.5 ± 8.1	Olfactory stimulation	4 weeks	CMAI
Lin et al. (2007) [39]	Hong Kong	RCT	n = 35, mean age 78.20 ± 3.79	Olfactory stimulation	3 weeks	NPI
Burns et al. (2011) [40]	UK	RCT	n = 32, mean age 85.6 ± 12.6	Olfactory stimulation	12 weeks	NPI
Sakamoto et al. (2012) [41]	Japan	RCT	n = 51, mean age 84.2 ± 7.8	Olfactory stimulation	12 months	CMAI
Fu, Moyle and Cooke (2013) [31]	Australia	RCT	n = 22, mean age 84.0 ± 6.36	Olfactory stimulation	6 weeks	CMAI
O’Connor et al. (2013) [42]	Australia	RCT	n = 37, mean age 77.6 ± 9.40	Olfactory stimulation	1 week	CMAI
Yang et al. (2015) [43]	Taiwan	RCT	n = 73, mean age 83.67 ± 4.96	Olfactory stimulation	4 weeks	CMAI
Takeda, Watanuki and Koyama (2017) [44]	Japan	Quasi-experimental design	n = 19, mean age 80.7± 9.1	Olfactory stimulation	20 days	NPI
Takahashi et al. (2020) [45]	Japan	RCT	n = 19, mean age 76.2 ± 9.8	Olfactory stimulation	8 weeks	NPI
Ting, Tien and Huang, (2023) [46]	Taiwan	Quasi-experimental design	n = 17, mean age 78.35 ± 13.35	Olfactory stimulation	4 weeks	CMAI
Burgio et al. (1996) [47]	USA	Quasi-experimental design	n = 13, mean age 83.08 ± 15.92	Nature sound	10 days	CMAI
Whall et al. (1997) [48]	USA	Quasi-experimental design	n = 15, mean age > 65	Nature image with sound	2 weeks	CMAI
Reynolds et al. (2018) [49]	USA	Quasi-experimental design	n = 14, mean age 85.1 ± 4.0	Nature image with sound	3 days	ABS
Ciobanu et al. (2022) [30]	Romania	Quasi-experimental design	n = 7, mean age 71.14 ± 12.17	Nature video with olfactory	4 weeks	CMAI
Lee and Kim (2008) [50]	Korea	Quasi-experimental design	n = 23, mean age > 65	Indoor garden	5 weeks	CMAI
Pedrinolla et al. (2019) [51]	Italy	RCT	n = 82, mean age 76.40 ± 4.30	Indoor garden	6 months	NPI
Wilkes et al. (2005) [52]	Australia	Quasi-experimental design	n = 16, mean age 79.86	Outdoor garden	3 months	CMAI
Detweiler et al. (2008) [53]	USA	Quasi-experimental design	n = 34, mean age 80.72 ± 6.72	Outdoor garden	12 months	CMAI
Murphy et al. (2010) [54]	USA	Quasi-experimental design	n = 34, mean age 84.2	Outdoor garden	12 months	CMAI
Luk et al. (2011) [55]	Hong Kong	RCT	n = 7, mean age 84.9 ± 8.30	Outdoor garden	6 weeks	CMAI
Edwards, McDonnell and Merl (2012) [56]	Australia	Quasi-experimental design	n = 10, mean age > 65	Outdoor garden	3 months	CMAI
Gueib et al. (2020) [7]	France	Quasi-experimental design	n = 16, mean age 82.1 ± 7.8	Outdoor garden	2 weeks	NPI
van der Velde-van Buuringen, Achterberg and Caljouw (2021) [8]	Netherlands	Quasi-experimental design	n = 20, mean age 85.2 ± 4.9	Outdoor garden	2 weeks	NPI
Motealleh et al. (2022) [9]	Australia	Quasi-experimental design	n = 10, mean age 81 ± 10.1	Outdoor garden	4 weeks	CMAI

Note: CMAI = Cohen–Mansfield Agitation Inventory, ABS = Agitated Behaviour Scale, NPI = Neuropsychiatric Inventory.

## Data Availability

No new data were created or analysed in this study.

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
