# Peer review of "Effectiveness of Nature-Based Interventions in Reducing Agitation Among Older Adults with Dementia: A Systematic Review and Meta-Analysis"

_healthcare, 2025, doi:10.3390/healthcare13141727_

Round 1
Reviewer 1 Report
Comments and Suggestions for Authors
This is a much needed topic and the methodology is well executed.
Could you please provide a brief justification for the use of the Meta-Essentials software package for data analysis? Specifically, it would be helpful to understand why this tool was selected over other commonly used meta-analysis software options and whether it offered particular advantages for the type of data or effect size calculations used in your review.
Could you also please clarify the rationale for estimating the correlation coefficient (r) from external sources rather than excluding those studies or using alternative imputation methods? It would also be helpful to specify the sources used for these estimations and the assumptions underlying this approach. This clarification is important for assessing the validity and reliability of the sensitivity analysis and overall findings.
The use of "we" is consistent across the paper; perhaps consider rephrasing this in the third person to align with academic conventions unless first-person usage is consistent with journal style guidelines. Thank you.
Author Response
Comments and Suggestions for Authors
Review 1:
This is a much needed topic and the methodology is well executed.
Could you please provide a brief justification for the use of the Meta-Essentials software package for data analysis? Specifically, it would be helpful to understand why this tool was selected over other commonly used meta-analysis software options and whether it offered particular advantages for the type of data or effect size calculations used in your review.
Response: Thank you for your thoughtful suggestion. We added a brief justification for the use of the Meta-Essentials software in 2.5. Data synthesis and analysis. “Meta Essentials is a useful tool that automatically calculate effect sizes from different types of data and supports various meta-analysis tasks, such as subgroup analysis, moderator analysis, and checking for publication bias.” (line 178-181)
Could you also please clarify the rationale for estimating the correlation coefficient (r) from external sources rather than excluding those studies or using alternative imputation methods? It would also be helpful to specify the sources used for these estimations and the assumptions underlying this approach. This clarification is important for assessing the validity and reliability of the sensitivity analysis and overall findings.
Response: Thank you for your insightful comment. As mentioned earlier, Meta Essentials is a valuable tool that automatically computes effect sizes from diverse statistical data. However, some studies did not provide sufficient statistical information, specifically the correlation coefficients needed for certain calculations.
Excluding these studies due to missing correlation coefficients would have led to a substantial loss of valuable data, potentially introducing bias and reducing the overall statistical power of our analysis. To mitigate this, we estimated the correlation coefficients (r) based on values reported in previously published meta-analyses and large-scale studies conducted in similar populations and contexts.
To ensure the robustness of our findings, we conducted sensitivity analyses by varying the estimated correlation coefficients within plausible ranges. This allowed us to assess the impact of these assumptions on the overall results.
The use of "we" is consistent across the paper; perhaps consider rephrasing this in the third person to align with academic conventions unless first-person usage is consistent with journal style guidelines.
Response: Thank you for your suggestion. We have revised the paper accordingly.
Reviewer 2 Report
Comments and Suggestions for Authors
To whom it may concern,
While the topic of the manuscript is of interest for a wide range of readers, the quality and structure need significant improvement before publication (major revision needed).
- Please change the title. I suggest you do not use a question. Also, you specifically state the type of paper (review/systematic review/meta-analysis).
- The abstract should be re-written according to the PRISMA recommendations if you want to write a meta-analysis.
- The introduction should comprise 4 paragraphs - the first three where you clearly state the existing knowledge and the rationale for the review. Moreover, the last paragraph should clearly state the aim and main points of your work.
- If you use abbreviations in the table, please add legends where you clearly state the meaning of these abbreviations. Also, in the title of the table/figure, add more details and the software used to create the figures (if any).
- The discussion section must be improved. Firstly, you should offer more details about about each intervention/setting (not as a whole group). Secondly, you should add some physiopathological explanations for your results.
- You must add a section/paragraph about the limitations of your study!
- The conclusion section should be re-written. Draw clearly 3-4 conclusion from your meta-analysis and suggest 2-3 future research directions.
- There is a high degree of similarity (30%). Try to rephrase that parts.
Author Response
Comments and Suggestions for Authors
Review 2:
While the topic of the manuscript is of interest for a wide range of readers, the quality and structure need significant improvement before publication (major revision needed).
Please change the title. I suggest you do not use a question. Also, you specifically state the type of paper (review/systematic review/meta-analysis).
Response: Thank you for your suggestion. We revised the title: “Effectiveness of nature-based interventions in reducing agitation among older adults with dementia: A systematic review and meta-analysis”
The abstract should be re-written according to the PRISMA recommendations if you want to write a meta-analysis.
Response: We have revised the abstract according to the PRISMA recommendations.
The introduction should comprise 4 paragraphs - the first three where you clearly state the existing knowledge and the rationale for the review. Moreover, the last paragraph should clearly state the aim and main points of your work.
Response: Thank you for your valuable feedback. I will revise the introduction to ensure it consists of four clear paragraphs.
If you use abbreviations in the table, please add legends where you clearly state the meaning of these abbreviations. Also, in the title of the table/figure, add more details and the software used to create the figures (if any).
Response: Thank you for your suggestion. We have added legends for Table 1.
The discussion section must be improved. Firstly, you should offer more details about each intervention/setting (not as a whole group). Secondly, you should add some physiopathological explanations for your results.
Response: Thank you for your valuable comments. However, due to the scope of the current study and the available data, we are unable to provide detailed physiopathological explanations at this time. Nevertheless, we will enhance the discussion by offering more detailed descriptions of each intervention and setting individually, as you suggested.
You must add a section/paragraph about the limitations of your study!
Response: Agree. We have added a section 4.3. Limitations of this study.
The conclusion section should be re-written. Draw clearly 3-4 conclusion from your meta-analysis and suggest 2-3 future research directions.
Response: Thank you for your suggestion. We have revised the conclusion accordingly.
There is a high degree of similarity (30%). Try to rephrase that parts.
Response: Thank you for your observation. We acknowledge that a significant portion of the similarity (30%) relates primarily to the description of methods and findings. We will carefully rephrase these sections to ensure originality while maintaining clarity and accuracy.
Reviewer 3 Report
Comments and Suggestions for Authors
Summary:
This study used PRISMA analyze 29 research papers and found that spending time in nature can help PLWD. The benefits were strongest when activities took place outdoors and included social interaction. It didn’t matter whether people had direct or indirect contact with nature, or how long the activities lasted. The results suggest that adding nature-based activities to care homes is a simple and effective way to help PLWD feel less agitated.
Comments:
Section 2.2:
I strongly recommend that the authors look for and include research about building density, building structure, and building direction.
Section 2.5:
Please give more details about the rules for including or leaving out studies. Also, clearly explain what you mean by ‘insufficient data,’ ‘incorrect data types,’ and ‘poor methodology’ in Figure 1.
Section 3.3:
Please explain why you removed two papers, which changed the total number from 31 to 29.
Section 3.4:
In the subgroup analysis, I suggest you compare the results of RCTs and quasi-experimental studies, since their methods and ways of choosing participants are different.
Section 3.4.1:
In Figure 2, many indoor results are important but very close to zero. Please check the scale and talk about this in the discussion section.
Section 3.4.2:
In Figure 3, many individual results are important but very close to zero. Please check the scale and talk about this in the discussion section.
Sections 3.4.3–3.4.4:
I suggest moving these sections to the appendix and discussing them separately after Section 4.2.
Author Response
Comments and Suggestions for Authors
Review 3:
This study used PRISMA analyze 29 research papers and found that spending time in nature can help PLWD. The benefits were strongest when activities took place outdoors and included social interaction. It didn’t matter whether people had direct or indirect contact with nature, or how long the activities lasted. The results suggest that adding nature-based activities to care homes is a simple and effective way to help PLWD feel less agitated.
Section 2.2:
I strongly recommend that the authors look for and include research about building density, building structure, and building direction.
Response: Thank you for your suggestion. Agree we attempted to identify the building density, building structure, and building orientation for the intervention settings in the studies. However, most studies did not provide detailed information. We have included this as a limitation of the study. (line 394-403)
Section 2.5:
Please give more details about the rules for including or leaving out studies. Also, clearly explain what you mean by ‘insufficient data,’ ‘incorrect data types,’ and ‘poor methodology’ in Figure 1.
Response: Thank you for your insightful comment. We have added the following paragraph to clarify the exclusion for readers: “A total of 57 studies underwent a full-text analysis. Studies that lacked essential statistical information (e.g., means, standard deviations, or sample sizes), presented data in incompatible formats, or demonstrated significant methodological flaws were excluded. This resulted in 31 articles in the final publication pool.” (line 226-230).
Section 3.3:
Please explain why you removed two papers, which changed the total number from 31 to 29.
Response: Thank you for your suggestions. We added the following sentence “Two studies were identified as outliers due to effect sizes substantially different from the others”. (line 263-264)
Section 3.4:
In the subgroup analysis, I suggest you compare the results of RCTs and quasi-experimental studies, since their methods and ways of choosing participants are different.
Response: Thank you for your suggestion. To systematically explore the relative impact of nature-based interventions on reducing residents’ agitation, we conducted subgroup analyses to investigate the factors affecting the effectiveness of natural contact and engagement on nature connection dimensions: (i) intervention settings (indoor vs. outdoor), (ii) the presence of social interaction, (iii) the types of experience with nature (indirect vs. direct) and (iv) the duration of interventions. Therefore, we are concerned that comparing the results of RCTs and quasi-experimental studies may lead to inconsistent findings. However, we have acknowledged the absence of sub-analysis results for RCTs and quasi-experimental studies as a limitation. (line 394-403)
Section 3.4.1:
In Figure 2, many indoor results are important but very close to zero. Please check the scale and talk about this in the discussion section.
Response: In Figure 2, effect size indicates the standardized difference between two means. This was followed by the evaluation of effect sizes, with 0.2, 0.5 and 0.8 representing small, medium and large effects, respectively. Although some studies showed small effects (very close to zero), we found that a large effect was observed (g = -1.00) for indoor settings, whilst a medium effect was found for outdoor settings (g = -0.76). we will enhance the discussion by offering more detailed descriptions of each intervention and setting individually, as you suggested. (line 341-345)
Section 3.4.2:
In Figure 3, many individual results are important but very close to zero. Please check the scale and talk about this in the discussion section.
Response:
Please see the above comment. Although some studies showed small effects (very close to zero), we found that individual interventions had a medium effect (g = -0.73), while interventions involving social interaction produced more significant outcomes (g = -0.99). we will enhance the discussion by offering more detailed descriptions of each intervention and setting individually, as you suggested. (line 369-374)
Sections 3.4.3–3.4.4:
I suggest moving these sections to the appendix and discussing them separately after Section 4.2.
Response:
Thank you for your suggestion. We conducted subgroup analyses to investigate the factors affecting the effectiveness of natural contact and engagement on nature connection dimensions: (i) intervention settings (indoor vs. outdoor), (ii) the presence of social interaction, (iii) the types of experience with nature (indirect vs. direct) and (iv) the duration of interventions. To improve clarity for readers, we believe the sections 3.4.3–3.4.4 should be placed under 3. Results 3.4. Subgroup analyses.
Round 2
Reviewer 2 Report
Comments and Suggestions for Authors
To whom it may concern,
Thank you for submitting the revised version of this manuscript. The authors have significantly improved the quality of the work and have responded to most of my comments. The following aspects remain to be clarified:
- "Due to the scope of the current study and the available data, we are unable to provide detailed physiopathological explanations at this time. Nevertheless, we will enhance the discussion by offering more detailed descriptions of each intervention and setting individually, as you suggested." Try to apply this principle to all the aspects included in the discussion section.
Author Response
Thank you for your feedback. We have applied this principle to all relevant aspects in the discussion section to improve clarity and depth. Please see line 369-371 & 390-401.
Reviewer 3 Report
Comments and Suggestions for Authors
NA
Author Response
Thank you for your supprt.